# A Compact Handheld Sensor Package with Sensor Fusion for Comprehensive and Robust 3D Mapping

**DOI:** 10.3390/s24082494

**Published:** 2024-04-12

**Authors:** Peng Wei, Kaiming Fu, Juan Villacres, Thomas Ke, Kay Krachenfels, Curtis Ryan Stofer, Nima Bayati, Qikai Gao, Bill Zhang, Eric Vanacker, Zhaodan Kong

**Affiliations:** 1Department of Biological and Agricultural Engineering, University of California, Davis, CA 95616, USA; penwei@ucdavis.edu (P.W.); jvillacres@ucdavis.edu (J.V.); 2Department of Electrical and Computer Engineering, University of California, Davis, CA 95616, USA; kmfu@ucdavis.edu; 3Department of Computer Science, University of California, Davis, CA 95616, USA; thke@ucdavis.edu (T.K.); kkrachenfels@ucdavis.edu (K.K.); crstofer@ucdavis.edu (C.R.S.); nbayati@ucdavis.edu (N.B.); 4Department of Mechanical and Aerospace Engineering, University of California, Davis, CA 95616, USA; qkgao@ucdavis.edu (Q.G.); bilzhang@ucdavis.edu (B.Z.); ecvanacker@ucdavis.edu (E.V.)

**Keywords:** 3D mapping, sensor fusion, SLAM, thermal camera

## Abstract

This paper introduces an innovative approach to 3D environmental mapping through the integration of a compact, handheld sensor package with a two-stage sensor fusion pipeline. The sensor package, incorporating LiDAR, IMU, RGB, and thermal cameras, enables comprehensive and robust 3D mapping of various environments. By leveraging Simultaneous Localization and Mapping (SLAM) and thermal imaging, our solution offers good performance in conditions where global positioning is unavailable and in visually degraded environments. The sensor package runs a real-time LiDAR-Inertial SLAM algorithm, generating a dense point cloud map that accurately reconstructs the geometric features of the environment. Following the acquisition of that point cloud, we post-process these data by fusing them with images from the RGB and thermal cameras and produce a detailed, color-enriched 3D map that is useful and adaptable to different mission requirements. We demonstrated our system in a variety of scenarios, from indoor to outdoor conditions, and the results showcased the effectiveness and applicability of our sensor package and fusion pipeline. This system can be applied in a wide range of applications, ranging from autonomous navigation to smart agriculture, and has the potential to make a substantial benefit across diverse fields.

## 1. Introduction

Mapping 3D environments has become an essential task across various fields. In robotics, for instance, constructing an accurate 3D map is crucial for the safe and efficient navigation of autonomous agents. These maps enable robots to understand and interact with their surroundings, facilitating tasks ranging from simple navigation to complex exploration of unknown environments [1]. Another application of advanced mapping technologies is in agriculture, where understanding the spatial distribution of fruits within tree canopies is essential for modern orchard management. Accurate 3D mapping aids in precise harvest forecasting, enabling meticulous planning and efficient resource allocation. Moreover, with the increasing automation in agriculture, robots designed for tasks such as fruit picking heavily rely on this spatial data. Detailed knowledge of fruit locations allows these robots to optimize their operations, minimizing fruit damage and ensuring efficient movements through orchards [2].

The rise of Simultaneous Localization and Mapping technology has greatly enhanced the ability to create a map of the environment in real-time while simultaneously tracking the agent’s location. This technology has found extensive applications in a variety of fields, such as autonomous driving [3] and augmented/virtual reality [4], and becomes extremely valuable in scenarios where global positioning (e.g., GPS) is not available. SLAM can rely on different sensing systems, including LiDAR and camera, each bringing unique strengths to the result. For example, LiDAR sensors generate accurate geometrical representations of the environment through dense point cloud, featuring a long range and high accuracy. However, they do not capture the visual appearance of objects, which can be crucial for recognizing and understanding the context of subjects within the mapped environments. Conversely, traditional cameras providing RGB images offer rich visual details but lack depth information, leading to an incomplete spatial understanding of the scene. The integration of LiDAR point cloud with camera pixel data thus offers a more comprehensive representation of the environment and benefits the 3D mapping and scene reconstruction, combining both geometric precision and visual context.

Existing mapping solutions often run into challenges in environments with compromised visual conditions, such as those with low illumination or the presence of airborne particulates (e.g., dust, fog, and smoke). Traditional RGB cameras may not function effectively in these scenarios. Alternative sensors like thermal-infrared cameras can provide valuable data that enable effective mapping even in challenging visual environments. In addition to the need for advanced sensing technology, there is also an increasing demand for a sensor package that is lightweight, compact, and easily portable. This combination would significantly improve the capability for robust and accurate 3D mapping across various scenarios.

In this work, we introduce a compact, handheld sensor package along with a sensor fusion pipeline aimed at enhancing 3D mapping by integrating multiple sensing technologies. By combining LiDAR, IMU, RGB camera, and thermal camera data, our solution provides a comprehensive and robust approach to environmental mapping, capable of functioning efficiently in diverse conditions and applications. Our sensor package, running a real-time LiDAR-inertial SLAM algorithm, allows for a dense and accurate geometric reconstruction of the environment. Concurrently, images are logged for post-processing. We fuse the generated point cloud map with images from the RGB or thermal camera in a post-processing approach, rendering it into a 3D map colored by RGB or thermal values. This method seeks to mitigate some of the challenges associated with existing mapping techniques, potentially enhancing exploration, navigation, and interaction in complex 3D environments. Our approach sets itself apart from existing handheld mapping solutions by integrating thermal imaging, which enables mapping in visually degraded conditions, a feature largely absent in current approaches. Furthermore, our comprehensive evaluation across varied environments, both indoor and outdoor, structured and unstructured, demonstrates the robustness and versatility of our system in real-world applications. The main contributions of the work are summarized as follows:We design a compact, handheld sensor package equipped with a 3D LiDAR, an RGB camera, a thermal camera, an IMU, and an onboard processing unit for comprehensive and robust 3D mapping.We propose a sensor fusion pipeline that integrates RGB or thermal images with a point cloud map produced by the LiDAR-Inertial SLAM algorithm to represent the 3D environments.We evaluate the performance of our handheld sensor package and sensor fusion pipeline in a variety of scenarios, demonstrating its efficacy in both indoor and outdoor settings, as well as in both structured and unstructured environments.We make our code available in a public repository (https://github.com/CHPS-Lab/handheld_mapping.git, (accessed on 7 April 2024)).

## 2. Related Work

Three-dimensional mapping has found applications in many fields. For example, it plays an important role in autonomous driving and drives technological advancements by creating a detailed environmental map [5]. In environmental monitoring and remote sensing, unmanned aerial vehicles (UAVs) can be employed for surveying and mapping to construct a map of the area of interest [6]. In agriculture settings, 3D mapping captures the fruit spatial distribution in orchards efficiently. While RGB cameras face issues like occlusions and low resolution in mapping fruit distribution [7], high-resolution LiDAR offers a detailed 3D point cloud of tree structures, partially overcoming these challenges. However, its high cost and the requirement to remain stationary during scans pose limitations [8]. Additionally, many of these mapping technologies rely on a global positioning system, which becomes problematic in environments where GPS signals are obstructed or unavailable. One such example includes scenarios like those encountered in the DARPA Subterranean Challenge [9], which demands exploring and mapping in unknown and GPS-denied environments. This necessitates the exploration of more comprehensive and robust mapping solutions.

State-of-the-art SLAM algorithms have demonstrated considerable success in a variety of applications. Extensive research into both camera-based [10,11] and LiDAR-based SLAM algorithms [12,13] has showcased their efficacy in a variety of scenarios. These algorithms can accurately localize robots while maintaining a detailed map of the environment in high resolution. The integration of inertial measurements has further improved the accuracy, efficiency, and reliability of these algorithms [14,15]. However, they encounter difficulties in adverse lighting conditions or in environments obscured by particulates, such as fog, dust, and smoke, where standard RGB cameras become ineffective. Thermal-infrared cameras, which detect surface radiation and estimate temperature based on the Stefan–Boltzmann equation, offer a robust alternative in such challenging conditions, unaffected by the limitation that impairs RGB cameras [16].

In the domain of 3D temperature mapping, thermal-infrared cameras have been extensively employed to capture detailed surface imagery of target objects, which provide richer information for mapping applications [17,18]. Nonetheless, in those works, they fused the thermal measurements with an RGBD camera’s output, which is limited in range and primarily suited for indoor environments. Recent research has explored the fusion of thermal measurements to enhance localization and mapping in broader visibility-reducing environments. For example, Chen et al. [19] introduced an RGB-T SLAM framework, merging RGB and thermal data for improved accuracy and robustness in varied lighting conditions. Shin and Kim [20] proposed a direct thermal-infrared SLAM algorithm utilizing sparse LiDAR output and demonstrated improved robustness under different illumination conditions. Saputra et al. [21] presented a probabilistic neural network noise abstraction method with robust pose graph optimization for a comprehensive thermal-infrared SLAM system. In addition, Khattak et al. [22] developed a keyframe-based thermal-inertial odometry algorithm for aerial robots, enabling navigation in GPS-denied and visually degraded environments. Polizzi et al. [23] devised a collaborative thermal-inertial odometry system, improving feature matching for loop closure detection and achieving efficient, decentralized state estimation for teams of flying robots. However, these methods mainly focus on integrating the thermal measurements to improve the robustness and accuracy of the state estimation and do not emphasize the generated 3D map by fusing different sources of sensors. A project closely aligned with our work is by Vidas et al. [17], who successfully implemented 3D radiometric mapping by combining data from LiDAR SLAM and a thermal camera.

Regarding other techniques for 3D map reconstruction, the Neural Radiance Field (NeRF) has recently emerged as a promising method which utilizes deep neural networks to reconstruct 3D scenes that can offer photorealistic rendering performance [24]. However, despite its potential, NeRF has the issue of scale ambiguity due to its inherent presence in the inferring process. Additionally, NeRF requires significant computational resources and time for training and rendering, which poses limitations for its deployment in scenarios demanding fast mapping.

In this work, we designed a compact handheld sensor package that enables fast and convenient 3D mapping of the surroundings using SLAM and sensor fusion. Many researchers have explored 3D mapping using handheld or mobile robot solutions. For example, Maset et al. [25] proposed a mapping system incorporating LiDAR and IMU data, evaluated in hand-carried and mobile platform-mounted configurations. However, this system does not include cameras and its performance in unstructured outdoor environments remains untested. Lewis et al. [26] studied collaborative 3D scene reconstruction in large outdoor environments using a fleet of mobile ground robots. Yet, their approach relies on GPS receivers for global localization, which might not be available in all scenarios. In addition, Ramezani et al. [27] provided a high-quality dataset comprising LiDAR, inertial, and visual data collected via a handheld device, alongside ground truth from a survey-grade LiDAR scanner, which has been used for benchmarking various algorithms. However, this dataset lacks thermal imaging. A similar device has been developed by Lin and Zhang [28], who used it to validate their LiDAR–Inertial–Visual sensor fusion algorithm. Nevertheless, their framework lacks support for thermal input, thus missing the capability to map in visually degraded environments.

## 3. Materials and Methods

### 3.1. Handheld Sensor Package

We built a custom handheld sensor package to balance performance with practicality and mobility, as shown in Figure 1. This sensor package is equipped with multiple sensors for comprehensive environmental mapping. At its core, it has a Velodyne Puck Lite 3D LiDAR (LiDAR Solutions, Edinburgh, VIC, Australia) for point cloud measuring and a VectorNav VN-100 IMU module (VectorNav, Dallas, TX, USA) to read acceleration and angular velocity data. We chose a Teledyne FLIR Vue Pro (Teledyne FLIR, Wilsonville, OR, USA) thermal camera for thermal imaging and an Intel RealSense D405 camera (Intel, Santa Clara, CA, USA) for RGB imaging (depth is not used). The specifications for each sensor are presented in Table 1. We installed an NVIDIA Jetson TX2 (Intel, Santa Clara, CA, USA) computer onboard, which processes all sensor measurements, runs the SLAM algorithm, and manages data logging. The software is running in the Robot Operating System (ROS). For immediate feedback on the mapping process, we included a portable LED screen for the user to visualize the real-time SLAM results when mapping the environment. We also added a pair of antennas to enable data sharing with a ground station, which is helpful for remote control and monitoring. The device is powered by a portable power bank, ensuring up to 3 h of continuous use. This power bank can be easily swapped out, which extends the devices’ use time during long-term missions. The total weight of the device is 2.4 kg. Designed for adaptability, our handheld sensor package can be easily extended with additional sensors. For example, one could add a compact air quality sensor to collect air samples simultaneously while mapping the environment, enhancing the device’s utility for advanced environmental analysis.

### 3.2. Calibration Procedure

To achieve accurate localization and mapping results, careful calibrations of the sensors are necessary. Note that for the Intel RealSense D405 camera, we treated it as a standard RGB camera and utilized only its RGB output. The depth feature was not used in our study. For the intrinsic calibration of the RGB camera, we utilized the Kalibr toolbox [29] and AprilTag markers [30] shown in Figure 2, which facilitate the determination of the intrinsic parameters (i.e., focal length, optical center) and distortion coefficients assuming a pinhole camera model. To calibrate the extrinsic parameters and time shift with respect to the IMU, we employed Kalibr’s visual–inertial calibration tool [31]. The calibration was conducted by moving our handheld sensor package in front of the AprilTag calibration board. We collected image and IMU data simultaneously from a variety of angles (roll/pitch/yaw) and distances (x/y/z directions) to sufficiently excite all degrees of freedom of the IMU. These data allow us to compare actual motion as recorded by the IMU against the motion inferred from camera images using the previously obtained intrinsic parameters and the calibration board. The tool runs a batch optimization algorithm to estimate both the extrinsic parameters and time offset between the two sensors. These procedures completed the intrinsic and extrinsic calibrations as well as the temporal alignment of the RGB camera.

Calibrating the FLIR Vue Pro (thermal) camera presented unique challenges, primarily due to the absence of a thermal gradient in AprilTag. To circumvent this, we designed a custom calibration board cut from a 3mm acrylic board, with a structured circle grid on it. This board was placed in front of a large metal plate. Prior to conducting intrinsic calibration, both the acrylic board and the metal plate were heated using a heat gun. The distinct thermal conductivity of the two materials allowed the holes to be distinctly visible through the thermal camera, as illustrated in Figure 3a. After capturing frames at different angles and distances, we employed a standard OpenCV library [32] to derive the thermal camera’s intrinsic parameters and distortion coefficients. The resulting undistorted image, demonstrating the calibration’s effectiveness, is displayed in Figure 3b. For the extrinsic calibration, we initially explored the approach presented in [33] that used a simple 3D geometry (e.g., a cardboard box) to estimate the transformation between the LiDAR and thermal camera. This approach aligns the thermal camera’s data with the LiDAR point cloud without the need for a target-recognizable calibration pattern. However, our experiments revealed challenges due to the sparse output from our Velodyne LiDAR—a spinning LiDAR emitting only 16 laser beams in contrast to the solid-state LiDAR used in the original paper—which lacks sufficient density for precise corner detection. Therefore, we adopted a similar visual–inertial calibration approach as the one used for the RGB camera. We used the Kalibr tool and a heated circular-pattern calibration board to estimate both the transformation and time shift between the thermal camera and IMU. The steps have been described above. These processes successfully calibrated the intrinsic and extrinsic as well as the time shift of our thermal camera in relation to the IMU.

The transformation and time offset between the 3D LiDAR and the IMU were calibrated using a targetless approach, as described by Lv et al. [34]. This method utilizes a continuous-time batch optimization approach to jointly minimize the IMU-based cost and LiDAR point-to-surfel distance on the data collected from LiDAR and IMU. In our system, we selected the position of IMU position as the origin. Consequently, we determined the transformation and time shift of all other sensors relative to the IMU. The detailed calibration results are presented in Table 2.

Although hardware-based synchronization methods, like the pulse per second (PPS) signal or network time protocol (NTP) method, could offer better accuracy, our handheld sensor package is designed for use in environments where both GPS signals and internet service may not be available. In addition, the RealSense D405 camera does not support external sensor synchronization. Given these constraints, we opted for a software-based approach to synchronize the sensors in our design.

### 3.3. LiDAR-Inertial SLAM

Matching image data with point cloud data to create a colored 3D map can present considerable challenges. While recent LiDAR–Inertial–Visual SLAM algorithms [28] have demonstrated some successes, fusing data from multiple sensor sources involves substantial processing and optimization. It may impose excessive computational demands on the onboard computing unit. Furthermore, the efficacy of RGB cameras can decrease significantly under visually degraded conditions. This limitation highlights the necessity for adding thermal imaging to achieve optimal performance in various operational scenarios. Our approach addresses these challenges by dividing the mission into two stages. During the first stage, we utilize LiDAR and IMU sensors along with a state-of-the-art LiDAR-Inertial SLAM algorithm to produce a dense point cloud map. The algorithm also captures the estimated poses at different frames. In the second stage, we integrate image data into the previously generated point cloud map through a series of post-processing steps. This two-stage approach offers a simple yet effective solution that maintains high accuracy without compromising computational efficiency.

In this work, we adopt the FAST-LIO2 framework introduced by Xu et al. [35] to achieve accurate localization and mapping of the surrounding environments using only LiDAR and IMU sensors. FAST-LIO2 stands out as a fast, robust, and accurate LiDAR-inertial mapping and odometry framework. In the FAST-LIO2 framework, the LiDAR motion distortion is effectively mitigated through the fusion of high-rate IMU measurements, ensuring precise alignment and accuracy of the spatial data. The introduced incremental k-d tree data structure delivers superior performance to existing algorithms while significantly reducing the computational time. It has demonstrated superior performance in terms of accuracy and computational efficiency when compared to other algorithms, including LIO-SAM, LINS, and LILI-OM [35], and is robust to aggressive motions and structureless environmental conditions. Moreover, the framework has been proven to operate effectively on both Intel-based and ARM-based CPUs. These advancements enable the algorithm to run efficiently on our Jetson TX2 computer, allowing for real-time computation and visualization of the localization and mapping results on the onboard LED screen.

Our TX2 computer is equipped with a quad-core Cortex-A57 processor, operating at a maximum of 2 GHz, and includes 8 GB of memory. On average, the total processing time for both mapping and odometry updates is less than 100 ms (occasional exceedances may occur). This processing time is well within the LiDAR’s sampling rate of 10 Hz during our tests. To alleviate the system’s load for visualization, we limited the update rate in ROS rviz to 1 Hz. To illustrate the effectiveness of the FAST-LIO2 algorithm, we show results from two distinct scenarios: an indoor hallway, depicted in Figure 4a, where the real-time localization and mapping results displayed on our handheld sensor package are shown in Figure 4b, and an outdoor semi-structured environment, displayed in Figure 5a. To further demonstrate the accuracy of this method, we overlay the generated point cloud map onto a satellite image from Google Maps, corresponding to the same outdoor area. This comparison, illustrated in Figure 5b, reveals a high degree of alignment between the point cloud map and the satellite image. These results highlight the robust performance of our LiDAR-Inertial SLAM algorithm under different conditions.

### 3.4. Point Cloud Rendering

We modified the original FAST-LIO2 code, so our LiDAR-Inertial SLAM output includes not only a dense point cloud map but also additional information, such as the count of newly scanned points, their estimated poses, and the timestamps of their registration. These supplementary data are crucial for correlating each point in the cloud with the corresponding image pixel. After acquiring the point cloud map generated by the SLAM algorithm, our subsequent step is to fuse image pixel values onto the corresponding points in the 3D space. This fusion is executed through a post-processing approach, designed to minimize the computational load on the onboard computer while providing us additional flexibility for filtering and optimization. During this post-processing process, we are able to choose the image source we want for rendering. We can produce a colored 3D map using either RGB camera data or thermal camera data, depending on the specific mission requirement. It is important to note that the version of the thermal camera we used does not provide radiometric data. Consequently, we are limited to obtaining temperature gradients from the camera’s output rather than precise temperature measurements for each pixel. However, given that our primary objective is to validate the effectiveness of our sensor fusion pipeline, we consider the current setup adequate for this purpose. In our framework, we process thermal images similarly to traditional RGB images. The input thermal image uses a green palette; each has a three-channel, 8-bit format.

The fusion process starts with extracting all image files from an ROS bag file with their pixel values and timestamps. Next, we apply preprocessing to both the images and point cloud data. Note that our thermal camera periodically undergoes an internal re-calibrating process, which causes frozen frames for a short time. During our handling of the thermal camera, we identify and drop these frames to ensure a reliable data input. After that, image distortion is corrected using the camera’s intrinsic parameters and distortion coefficients, resulting in undistorted images. For the point cloud, we estimate normals based on the local density and refine the cloud by aligning normals towards the data capture position, leveraging the known poses and registered timestamps from our modified SLAM algorithm. This step helps in removing ambiguity in orientation and filtering out potentially problematic points.

Subsequently, we employ a virtual depth camera, constructed through extrinsic parameters between the IMU and the corresponding camera, to project views onto the processed point cloud. Camera poses corresponding to image timestamps are interpolated from the logged trajectory using dual-quaternion linear blending [36]. Suppose we have two dual-quaterions ${\hat{Q}}_{1}$ and ${\hat{Q}}_{2}$; the interpolated dual-quaternion ${\hat{Q}}_{interp}$ is calculated as:(1)$${\hat{Q}}_{interp}=\frac{\left(1-a\right){\hat{Q}}_{1}+a{\hat{Q}}_{2}}{∥\left(1-a\right){\hat{Q}}_{1}+a{\hat{Q}}_{2}∥}$$
where *a* is an interpolation factor which can be calculated from the timestamps of ${\hat{Q}}_{1}\left(t\right)$, ${\hat{Q}}_{2}\left(t\right)$, and ${\hat{Q}}_{interp}\left(t\right)$ data. For efficiency, the depth is limited to 10 m. With this virtual depth camera output, we combine depth and RGB data into RGBD images, which are then reprojected into a colored map via volumetric integration. For this task, we utilize Open3D’s truncated signed distance function (TSDF) [37]. TSDF represents a 3D voxel array where each voxel stores the truncated signed distance to the nearest surface, effectively capturing objects within a volume of space. This structure enables TSDF to integrate multiple observations, functioning like a weighted average filter in 3D space to smooth out noise and discrepancies. This allows us to reconstruct a dense, colored point cloud map. To balance the quality and processing efficiency, we chose a voxel size of 0.01 m. In the final step, we apply statistical outlier removal to clean the point cloud and optionally downsample the point cloud to manage the data volume, yielding the desired output. The workflow of our 3D mapping and sensor fusion pipeline is depicted in Figure 6.

## 4. Experimental Results

### 4.1. Results on Indoor Tests

We now present the experiments we performed and their corresponding results. The first experiment we conducted was to assess the performance of our framework in an indoor setting focusing on 3D mapping with a thermal camera. The utilization of thermal imaging for 3D mapping is particularly valuable in scenarios such as search and rescue operations. The ability to create 3D thermal maps can aid in quickly locating individuals in smoke-filled or low-visibility environments by detecting heat signatures. To simulate subjects with varying temperatures, we placed a cardboard box containing ice in a hallway and surrounded it with hand warmers. Utilizing our handheld sensor package, we first scanned the hallway to produce a 3D point cloud map, as illustrated in Figure 7a. Meanwhile, we also recorded the thermal images streamed from the camera, shown in Figure 7b. As described in Section 3.4, we first corrected the images for distortion using the camera’s intrinsic parameters and distortion coefficients and also filtered the point cloud. We then established a virtual depth camera view by merging the history camera poses with point cloud data in Figure 7c. This setup enabled us to combine the images with depth information into RGBD images and subsequently reproject these RGBD images into a colored map using TSDF, as depicted in Figure 7d. In the final step of post-processing, we employed a ’seismic’ colormap to visually represent the final 3D map, displayed in Figure 7e, where blue indicates cooler temperatures and red indicates warmer areas. This visualization distinctly highlights both the ice-filled box and the hand warmers, and even the location of hallway lights, identifiable by their heat emissions. These results demonstrate the effectiveness of our 3D thermal mapping and sensor fusion pipeline.

In addition to our thermal imaging experiments, we conducted another test to evaluate the performance using the RGB camera. This experiment focused on 3D map reconstruction in an indoor setting containing an artificial apple tree. Figure 8a displays the artificial apple tree positioned within a room. Employing our handheld sensor package, which executed the LiDAR-Inertial SLAM algorithm onboard, we walked around the room and collected data. This process yielded a dense point cloud map of the environment, including the apple tree, as illustrated in Figure 8b. Leveraging the point cloud rendering framework described earlier, we generated a 3D RGB map of the scene, shown in Figure 8c. Furthermore, we manually identified and marked the locations of the apples on the 3D map, with the results showcased in Figure 8d. These findings confirm that our handheld sensor package, coupled with the SLAM and sensor fusion pipeline, provides an effective tool for both fruit localization and environmental reconstruction tasks.

We also reported quantitative results for the processed point cloud and reconstructed 3D map from the indoor apple tree experiment. The point cloud rendering was executed on a computer equipped with an AMD Ryzen 9 3900X CPU and 64 GB of memory. The LiDAR-Inertial algorithm generated a point cloud map containing a total of 18,821,342 points, resulting in a file size of 451.7 MB. From the dataset, we generated 1176 virtual RGBD images. With a voxel size set to 0.01 m, the final reconstructed 3D map comprised 5,967,602 points and was 167.1 MB in size. The processing time was approximately 9 min. Adjusting the voxel size to 0.005 m increased the final reconstructed 3D map to 28,215,023 points and 790.0 MB size, with the processing duration extending to 15 min. Note that these computations were performed solely on the CPU without GPU acceleration, due to a compatibility issue with the Open3D library. A key direction for our future work is to leverage GPU acceleration, which we anticipate will significantly reduce the processing time.

### 4.2. Results on Outdoor Tests

We extended our testing to outdoor environments, starting with a parking lot scenario. This test was conducted to assess our system’s effectiveness in thermal rendering under more complex and dynamic conditions than those encountered indoors. Following the established procedure, we generated a 3D thermal map, as depicted in Figure 9. It is evident that the cars, having been exposed to sunlight (data were collected in the afternoon), exhibit significantly higher surface temperatures compared to their surroundings. Similarly, the building’s surface, having absorbed heat, also shows higher temperatures. This result effectively demonstrates our system’s capability in outdoor semi-structured environments, highlighting its efficacy in accurately capturing thermal variations.

An additional experiment was conducted to assess our system’s efficacy within agricultural contexts. This setting presents complex challenges due to its unstructured nature and environmental disturbances. We focused on evaluating the RGB camera’s performance through two separate tests: one in a peach orchard and another in a walnut orchard. In addition, for the peach orchard, we also collected the point cloud with the RIEGL VZ-1000 (Riegl USA, Orlando, FL, USA) high-resolution LiDAR for benchmarking purposes. The RIEGL VZ-1000 was placed in four positions, forming a square surrounding each peach tree. The individual scans were co-registered using the proprietary software RiScan PRO. For each experiment, we targeted a specific segment of a tree row for data collection and processing through our sensor fusion pipeline. The outcomes of these experiments are illustrated in Figure 10. The results obtained demonstrate the capability of our framework to deliver detailed and fairly accurate representations of agricultural scenes. Notably, the map precisely captures the foliage and tree trunks, with even ground details such as grass and shadows being distinctly rendered. However, it is important to acknowledge certain challenges we faced during the generation of these results. One primary challenge is attributed to the height of the trees. Since the scanning was conducted by a person, reaching the upper sections of the trees to scan them comprehensively proved difficult. This leads to a partial absence of data for higher areas. Furthermore, some parts of the trees are particularly thin, resulting in sparse and lower quality in those regions. Consequently, these portions might blend into the sky in the background due to the lack of detailed information. Nevertheless, despite these challenges, the overall quality of the generated map is acceptable, highlighting the potential of our system to provide valuable insights into agricultural settings. To further assess the quality of the map generated, we compared the similarity of the point cloud obtained with our system against the point cloud recorded by the high-resolution RIEGL VZ-1000 (see Figure 11a). The two point clouds are presented in the Figure 11b. The similarity was evaluated using the metric Average Ratio (AR) proposed by Berens et al. [38] and presented in Equation (Equation 2). This method compares two point clouds by using multiple thresholds and ensuring that both point clouds are equally considered.
(2)$$\begin{array}{c} \mathrm{AR}\left(X,Y\right):=\frac{\sum_{i=1}^{N}i\frac{|S_{D_{i}X,Y}|}{\left|X\right|}+\sum_{i=1}^{N}i\frac{|S_{D_{i}Y,X}|}{\left|Y\right|}}{N^{2}+N} \end{array}$$
where *X* and *Y* are the points in the first and second point cloud, respectively. $S_{D_{i}X,Y}$ represents the set of points in *X* that have a distance smaller than $D_{i}$ to one point in *Y*. This metric considers *N* thresholds (*D*) to assess the point cloud similarity. Following the guidelines presented in Berens et al. [38], we computed the AR between the point cloud obtained with the RIEGL LiDAR (i.e., ground-truth) and the point cloud obtained using our device. The result obtained for the peach data was AR = 0.89; it is worth noting that a perfect match between the two point clouds would be equal to 1.

## 5. Discussion

While there are studies focusing on developing real-time comprehensive LiDAR–Inertial–Visual SLAM frameworks, our work distinguishes itself by adopting a two-stage approach, with a significant emphasis on post-processing for 3D map construction. This framework not only offers enhanced flexibility for optimizing map quality, but also reduces the burden placed on the onboard processing unit. Meanwhile, it provides users with intermediate information, which can be crucial for mission success. In addition, the incorporation of thermal imaging on our sensor package is particularly useful under scenarios where low illumination or environmental factors such as fog, smoke, or darkness are present, enabling the detection and differentiation of objects based on their heat signatures. In our two-stage approach, we first leverage a proven SLAM algorithm to address the challenges of localization and mapping, while the user has access to monitoring these intermediate results. In the second stage, we integrate RGB or thermal images, depending on the mission requirement, with the pre-established point cloud via a carefully designed post-processing sequence. This results in a comprehensive and robust reconstruction of the environment.

We emphasize the significance of 3D mapping for agriculture, particularly for improving fruit picking and orchard management. In this work, we seek to provide a solution using the handheld sensor package we designed. This sensor package, combining the precision of LiDAR sensors with informative camera outputs, addresses the limitations of individual sensing sources and other challenges, such as view occlusions, low image quality, limited mobility, and unreliable GPS signal under tree canopies. Our experiments demonstrate that our sensor package and algorithm can accurately create 3D maps in these scenarios. In particular, our artificial apple tree experiment shows that we could locate the fruit position precisely, though it currently requires human labeling. We could enhance this process by leveraging state-of-the-art computer vision techniques, for instance, employing a re-trained YOLO network for automated fruit detection and labeling directly within our software framework. However, this advancement is beyond the scope of this paper. Implementing such automatic 3D fruit detection based on our generated 3D map promises to significantly benefit harvesting operations. Moreover, the digital models that we generated from various tree configurations and fruit distributions under real-world conditions could serve as valuable datasets for future research work.

A potential enhancement for our work is to refine the data structure of our color-rendered maps. Currently utilizing the TSDF method, which relies on a voxel map, we have identified memory capacity issues when generating results for large-scale environments. A promising improvement could involve the adoption of an OcTree data structure [39], which offers a more memory-efficient approach to spatial division. We would also like to upgrade our thermal camera to a radiometric model. Therefore, we can obtain real temperature measurements for each pixel. Additionally, our handheld sensor package’s considerable weight poses challenges for long-term use. We are in the process of developing a backpack variant that relocates some components into a backpack, significantly alleviating the burden on the user.

Our handheld device serves as a proof-of-concept of our mapping and sensor fusion techniques. In the future, we could adapt the core components of our sensor package for deployment on mobile robots (e.g., wheeled robots, UAVs, legged robots) and facilitate applications in challenging scenarios such as autonomous mapping and navigation. This system has the potential to navigate autonomously in areas inaccessible to humans, such as tunnels, forests, mines, and caves, and offers the capability to perform autonomous environmental mapping. Robots equipped with our technology could deliver detailed environmental representations while mitigating the risks humans face when entering such hazardous places.

## 6. Conclusions

In conclusion, our work introduces a novel handheld sensor package and sensor fusion pipeline that significantly advances the capabilities for 3D environmental mapping. By integrating data from LiDAR, IMU, RGB, and thermal cameras, we overcome limitations faced by existing mapping solutions, especially in environments with challenging visual conditions and unreliable global positioning. Our system adopts a two-stage sensor fusion framework. First, we create an accurate point cloud map of the environment using a proven LiDAR-Inertial SLAM algorithm, and later, we render the point cloud map using camera data to enhance the visual representation of the mapped areas in a post-processing step. The successful deployment of our system in both indoor and outdoor environments, through structured and unstructured scenarios, highlights its adaptability and efficacy. Our system can be applied in a broad range of applications, from autonomous robotics to smart agriculture. Furthermore, we make our code open-source and hope to contribute to the ongoing growth and development of 3D mapping technologies.

## Figures and Tables

**Figure 1 sensors-24-02494-f001:**
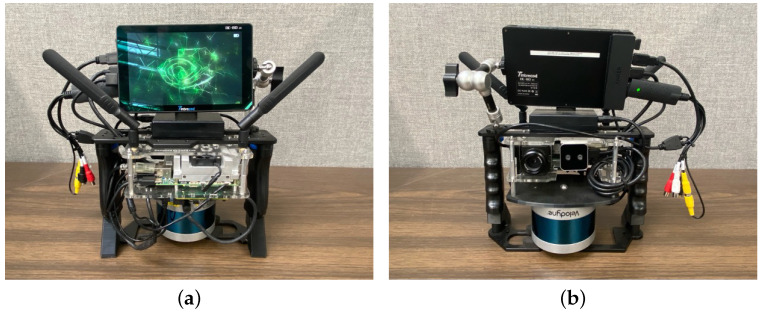
(**a**) Front view and (**b**) rear view of our handheld sensor package.

**Figure 2 sensors-24-02494-f002:**
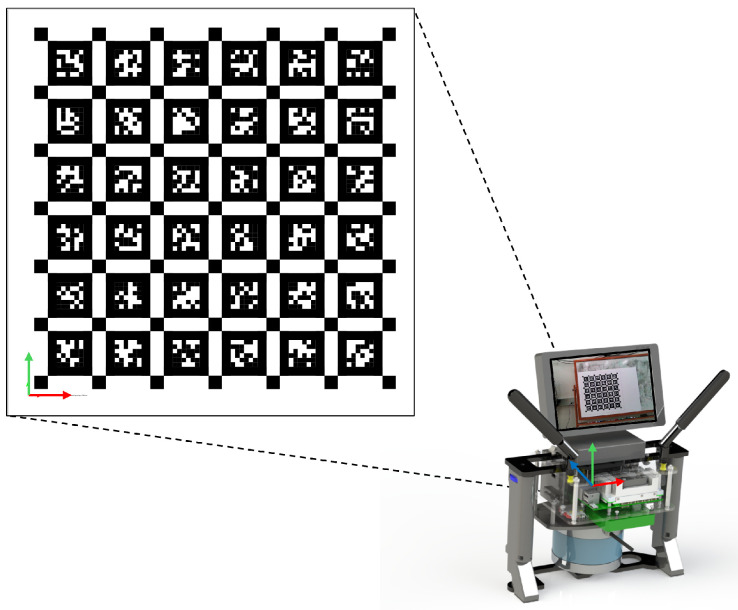
Calibration of RGB camera’s intrinsic and extrinsic parameters using AprilTag fiducial marker.

**Figure 3 sensors-24-02494-f003:**
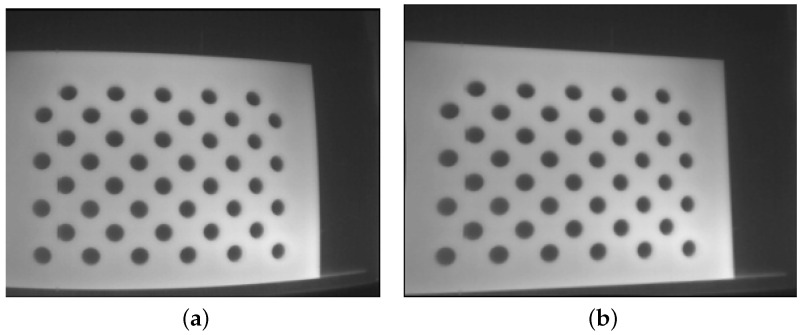
We used a custom board to calibrate the thermal camera. The board was heated during the calibration. (**a**) is the original thermal image, and (**b**) is the undistorted thermal image after we applied the identified intrinsic parameters and distortion coefficients of the camera.

**Figure 4 sensors-24-02494-f004:**
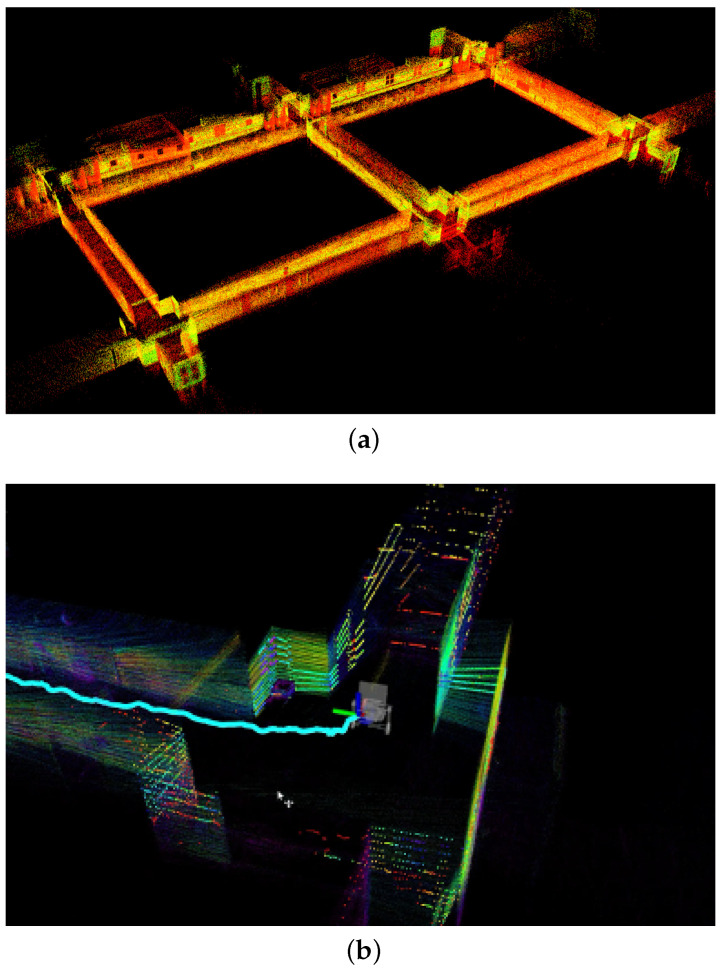
(**a**) LiDAR-Inertial SLAM performance in a hallway; (**b**) real-time localization and mapping results displayed on the handheld sensor package.

**Figure 5 sensors-24-02494-f005:**
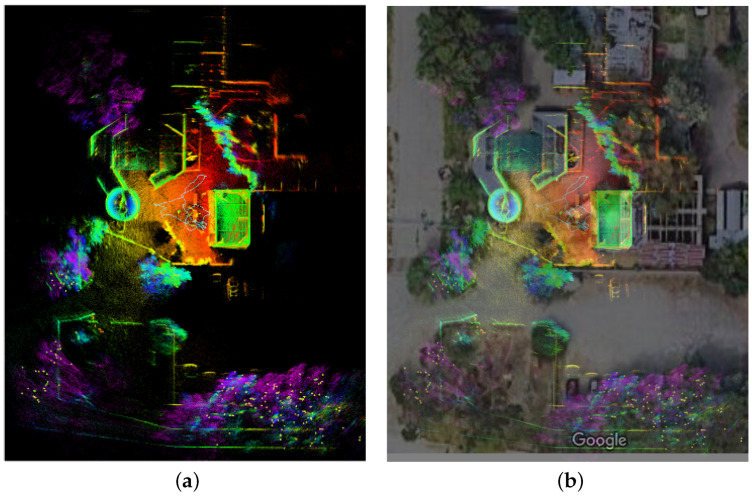
LiDAR-Inertial SLAM performance in an outdoor environment, where (**a**) is the generated point cloud map in a bird’s eye view, and (**b**) is the same map overlapped with a satellite image of the same area.

**Figure 6 sensors-24-02494-f006:**
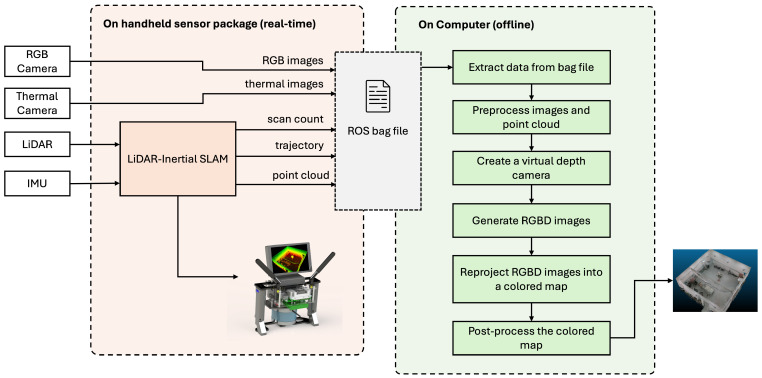
Overview of the system.

**Figure 7 sensors-24-02494-f007:**
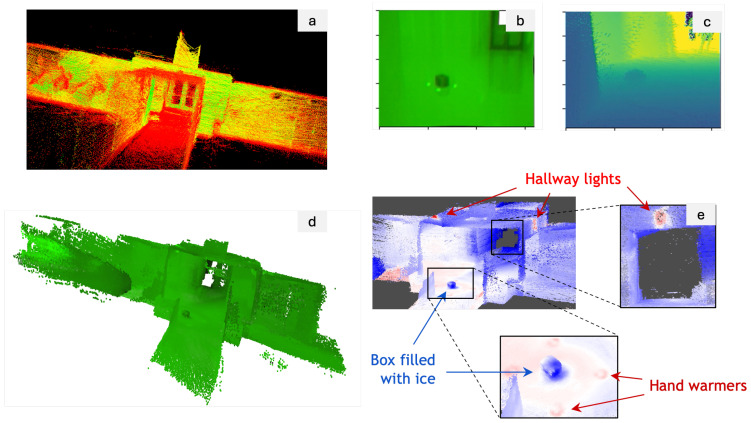
Results from the indoor 3D thermal map reconstruction: (**a**) SLAM-generated point cloud map, (**b**) thermal image, (**c**) virtual depth camera view, (**d**) colored 3D map reprojected from the RGBD images, and (**e**) the final visualization of the 3D map.

**Figure 8 sensors-24-02494-f008:**
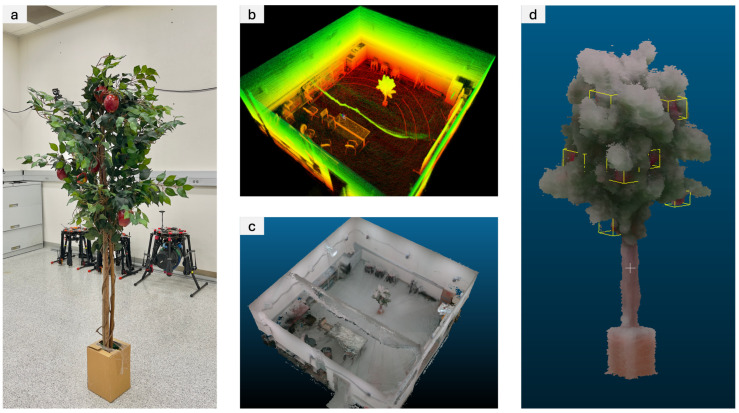
Results from the indoor 3D RGB map reconstruction: (**a**) photograph of an artificial apple tree, (**b**) point cloud generated by the SLAM algorithm, (**c**) RGB-rendered map after fusing RGB camera data, and (**d**) cropped view of the reconstructed apple tree with manually labeled apple positions.

**Figure 9 sensors-24-02494-f009:**
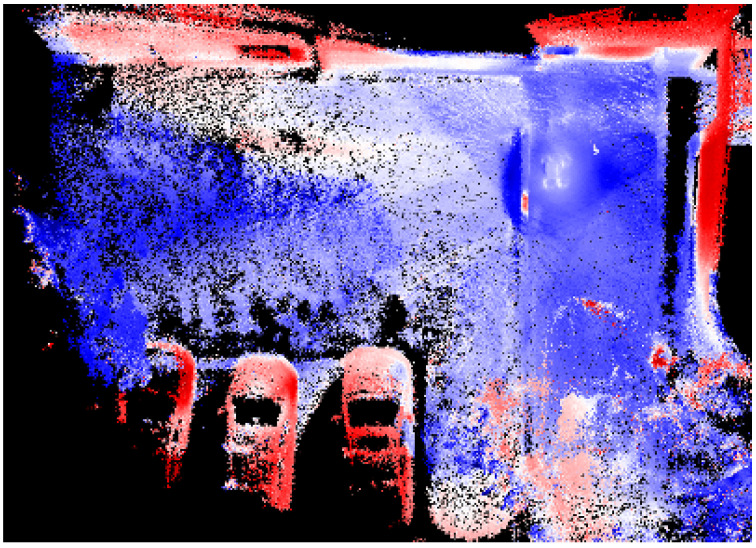
Visualization of the 3D thermal map generated from an outdoor parking lot.

**Figure 10 sensors-24-02494-f010:**
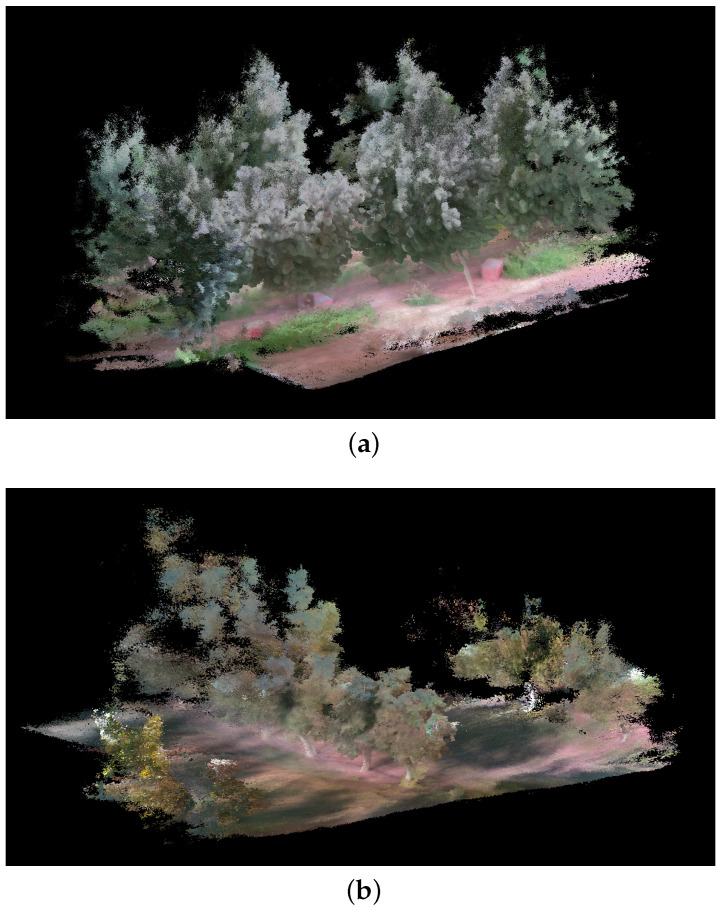
Three-dimensional map reconstruction results in (**a**) a peach orchard and (**b**) a walnut orchard.

**Figure 11 sensors-24-02494-f011:**
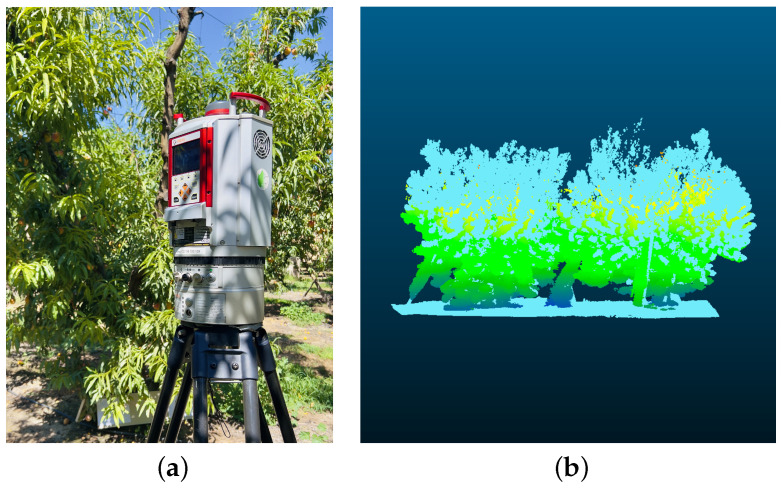
Point cloud similarity comparison: (**a**) ground-truth data captured with the high-resolution RIEGL VZ-1000 3D Terrestrial Laser Scanner, (**b**) overlay of registered point clouds, with RIEGL VZ-1000 data in light blue and our system’s derived point cloud in light green.

**Table 1 sensors-24-02494-t001:** Main sensors and their specifications.

Sensor	Specifications
Velodyne Puck Lite 3D LiDAR(Velodyne Lidar, Inc., San Jose, CA, USA) 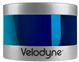	Number of channels: 16Maximum measurement range: 100 mFOV (vertical): ±15°FOV (horizontal): 360°Frame rate: 10 HzWeight: 600 gPower consumption: 8 W
Teledyne FLIR Vue Pro Thermal Camera(FLIR Systems, Inc., Wilsonville, OR, USA) 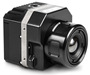	Resolution: 336×256FOV: 44°×33°Lens: 6.8 mmSpectral band: 7.5–13.5 µmFrame rate: 30 HzWeight: 113.4 gPower consumption: 2.1 W
Intel RealSense D405 RGBD Camera(Intel, Santa Clara, CA, USA) 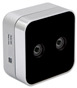	Resolution: Up to 1280×720FOV: 87°×58°Frame rate: 30 HzWeight: 60 gPower consumption: 1.55 W
VectorNav VN-100 IMU(VectorNav, Dallas, TX, USA) 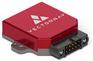	Accel. range: ±15 gAccel. bias stability: <0.04 mgGyro. range: ±2000°/sGyro. bias stability: 5–7°/hFrame rate: 400 HzWeight: 15 gPower consumption: 220 mW

**Table 2 sensors-24-02494-t002:** Calibration results of different sensors.

Parameter	Value
RGB Camera:	
focal length	$f_{x}=395.88935644$, $f_{y}=396.58689448$
optical center	$c_{x}=321.40957052$, $c_{y}=237.46367861$
distortion coefficients 1	$k_{1}=-0.0455943922$, $k_{2}=0.0397782471$
distortion coefficients 2	$p_{1}=-0.00179379601$, $p_{2}=-0.000123883618$
extrinsic translation (m)	xyz=[0.031094,−0.024541,0.031936]
extrinsic rotation (rad)	rpy=[−1.5589,0.0176,−1.5601]
time shift to IMU (s)	Δt=0.017
Thermal Camera:	
focal length	$f_{x}=413.42478787$, $f_{y}=411.95068373$
optical center	$c_{x}=159.97522005$, $c_{y}=121.32820803$
distortion coefficients 1	$k_{1}=-0.424659888$, $k_{2}=0.501250141$
distortion coefficients 2	$p_{1}=-0.00250397308$, $p_{2}=0.000725460516$
extrinsic translation (m)	xyz=[0.054164,−0.091582,0.031582]
extrinsic rotation (rad)	rpy=[−1.5708,0.0083,1.5708]
time shift to IMU (s)	Δt=0.100
LiDAR:	
extrinsic translation (m)	xyz=[0.039944,−0.051082,−0.06545]
extrinsic rotation (rad)	rpy=[−3.1416,0.0107,0.0014]
time shift to IMU (s)	Δt=0.005
IMU:	
accelerometer noise density (m/s^2^/$\sqrt{\mathrm{Hz}}$)	$\sigma_{accel}=0.01418$
accelerometer random walk (m/s^3^/$\sqrt{\mathrm{Hz}}$)	$\beta_{accel}=0.0004978$
gyroscope noise density (rad/s/$\sqrt{\mathrm{Hz}}$)	$\sigma_{gyro}=0.0008037$
gyroscope random walk (rad/s^2^/$\sqrt{\mathrm{Hz}}$)	$\beta_{gyro}=0.000006203$

## Data Availability

Data are contained within the article.

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
