# Peer review of "A Compact Handheld Sensor Package with Sensor Fusion for Comprehensive and Robust 3D Mapping"

_sensors, 2024, doi:10.3390/s24082494_

Round 1

Reviewer 1 Report

Comments and Suggestions for Authors 1. In the hardward design part, did you sync all the senesors like using PPS signal or NTP method, if not please give a reason? 2.Is that the RealSense RGBD Camera only provides the RGB, the depth is not shown in Figure 6? 3.In the Table 2, you mentioned the time shift, how to get the time shift between different sensors? 4. Please give a time useage on TX2, what is the time consumption of some opensource LiDAR-Visial-Inertial algorithm running time on TX2? 5.Figure 7 and Figure 9, when you use thermal point cloud, please add a color to temperature bar. 6.When you add the color to the point cloud, did you consider the LiDAR motion distortion? 7.Why FAST-LIO2? maybe should compares with other algorithm like LIOSAM in accuracy and realtime behaviour. Comments on the Quality of English Language

The Quality of the English Language is fine.

Reviewer 2 Report

Comments and Suggestions for Authors

The paper presents a compact handheld sensor package with sensor fusion for comprehensive and robust 3D mapping. The topic of the paper is interesting. The manuscript is overall well written, easy to read and properly organized. I have the following comments to improve the quality of the paper:

·         The main contributions of the paper should be better compared with respect to similar works on the topic of 3D mapping using handheld solutions.

·         More details on the LiDAR-thermal camera calibration should be added to the paper.

·         I suggest adding more information about the experimental results, as for instance, quantitative data on the processed point clouds and 3D reconstruction (number of points, size of data in MB or GB, point density, computational processing time, etc.). It would also be interesting to show the path of the sensor package during the acquisitions.

·         It would be interesting to evaluate the accuracy of the reconstruction, for example by comparing the experimental results with a ground truth (compared to alternative strategies).

·         It would also be interesting to discuss the possible applicability of the proposed solution to autonomous mapping using a mobile robot or an UAV.

·         The literature review should be improved by considering additional works on the topic. See for instance:

o   (2022). Performance investigation and repeatability assessment of a mobile robotic system for 3D mapping. Robotics, 11(3), 54.

o   (2022). Collaborative 3D scene reconstruction in large outdoor environments using a fleet of mobile ground robots. Sensors, 23(1), 375.

o   (2020). The newer college dataset: Handheld lidar, inertial and vision with ground truth. In 2020 IEEE/RSJ International Conference on Intelligent Robots and Systems (IROS) (pp. 4353-4360). IEEE.

Round 2

Reviewer 1 Report

Comments and Suggestions for Authors

1.Please add more experiments results compares with the Riegl ground truth.

Reviewer 2 Report

Comments and Suggestions for Authors

The paper has been improved by following all my previous comments. I suggest the paper to be accepted for publication.

Author Response

Thank you very much!